# A study on the measurement of inter-provincial trade costs in Yangtze River Delta from the perspective of value-added trade and its promotion effect

Renxiang Lu[1], Yang Wang[1]*, Huanyue Tong[2]

1 School of Business, Shanghai Dianji University, Shanghai, China, 2 Shanghai Wusongkou Culture & Tourism Investment (Group) Co., Ltd., Shanghai, China

* yangwang@sdju.edu.cn

**Data Availability Statement:** All data files are available from the Input-output table between provinces and cities in China.

## Abstract

This article uses the Novy model improved by value-added trade data to measure the cost of inter-provincial trade in the Yangtze River Delta and, on this basis, uses the differential decomposition method to explore the promoting effect of inter-provincial trade costs on the development of inter-provincial trade in the Yangtze River Delta. The results show that the inter-provincial trade costs of the provinces/municipalities in the Yangtze River Delta have increased and decreased, but the changes are small, and there are significant differences in sectoral and bilateral trade costs; the results of the differential decomposition show that the contribution of inter-provincial trade costs to inter-provincial trade development in the Yangtze River Delta is much lower than that of economic growth. Therefore, the Yangtze River Delta should deepen the division of labor and cooperation, give full play to the radiation and leading role of Shanghai as an economic center, and accelerate the digital innovation transformation of the service industry to reduce inter-provincial trade costs and achieve a higher level of integrated development of the Yangtze River Delta.

## 1. Introduction

The Yangtze River Delta (YRD) region has been one of the most dynamic regions in China in terms of economic development, openness, and innovation, as well as the intersection of the "One Belt, One Road" and the Yangtze River Economic Belt, which is strategically vital for China's modernization and all-round opening up to the outside world. The General Secretary announced at the first China International Import Expo that the Yangtze River Delta region would be supported and elevated to a national strategy, marking a new stage of development for the Yangtze River Delta region. The Outline of the 14th Five-Year Plan and 2035 Vision for National Economic and Social Development of the People's Republic of China, released in 2021, once again reaffirmed that the Yangtze River Delta is the first place to accelerate the creation of a leading high-quality development echelon, aiming at international advanced science and innovation capabilities and industrial systems, accelerating the G60 science and

**Funding:** The author(s) received no specific funding for this work.

**Competing interests:** The authors have declared that no competing interests exist.

innovation corridor of the Yangtze River Delta and the industrial innovation belt along Shanghai and Nanjing, improving the global resource allocation capability of the Yangtze River Delta and its radiation-driven ability to the whole country, and striving to enhance the level of integrated development of the Yangtze River Delta.

After years of grinding development, the comprehensive development level and synergy show that the Yangtze River Delta region has formed a better synergistic development situation [1]. Since the Yangtze River Delta integration was upgraded to a national strategy in November 2018, the GDP of the Yangtze River Delta region has grown from 22.1 trillion yuan in 2018 to 27.6 trillion yuan in 2021. the per capita GDP of the Yangtze River Delta region reached 117,300 yuan in 2021, which is already nearly twice the level of GDP per capita of middle and high-income countries in 2020 [2]. However, compared with Beijing-Tianjin-Hebei and the Pearl River Delta, the level of integrated development and market integration in the Yangtze River Delta is still low, the industrial layout is unreasonable, and the pressure for transformation and upgrading is high [3–5]. Factor markets' integration level lags far behind commodity markets [6]. In addition to economic factors, various institutional barriers contribute to the distorted development of YRD integration [7]. The uneven development within the Yangtze River Delta region, the severe phenomenon of "vassal economy," the stagnation of market integration and industrial division of labor, the difficulty of synergistic development between mega-cities and super-small cities, the inadequate coordination mechanism of regional interests and the current governmental assessment system, and the failure of Shanghai's core leading role to give full play to the Yangtze River Delta integration are all severe constraints to the development level [8–10]. Therefore, only by optimizing the industrial layout, strengthening the coordination of policies and interests within the region, deepening the division of labor and cooperation, and enhancing the capacity for collaborative scientific and technological innovation can we achieve a higher level of integrated development in the Yangtze River Delta [11–13].

Given the economic development level, industrial layout, technology level, and market scale of the Yangtze River Delta, the Yangtze River Delta region can fully realize the division of labor and complementary advantages of industries by reducing inter-provincial trade costs and developing inter-provincial trade to enhance the level of integrated development. For a long time, trade costs have played an essential role in the specialized division of labor and cooperation, becoming the key to explaining the location selection and economic activities of the division of labor entities participating in the division of labor and cooperation activities. The mechanism of the impact of trade costs on the division of labor and cooperation can be summarized as creation effects and transfer effects. The reduction of trade costs in a particular region means that the division of labor and trade in that region is more convenient, which will attract many enterprises to enter the area and form industrial clusters, resulting in the new division of labor cooperation and creating a creative effect of the division of labor cooperation. In addition, as the cost of trade in a specific region decreases and business becomes more convenient, some or even all the external division of labor and cooperation between enterprises within the region and those outside the area will be transferred to enterprises within the same region, resulting in a decrease in costs and an increase in efficiency. This is not the transfer effect of trade cost changes in the division of labor and cooperation. From this, it is not very objective to study the inter-provincial division of labor cooperation and its integrated development in the Yangtze River Delta without considering trade costs in the context of value chain division of labor cooperation. Because of this, as a critical factor affecting the inter-provincial division of labor and cooperation in the Yangtze River Delta, the actual situation and evolution trend of inter-provincial trade costs have become an important starting point for research. Therefore, this article intends to measure the cost of inter-provincial trade in the Yangtze

River Delta, examine the evolution trend of inter-provincial trade costs and their promoting effects on the development of inter-provincial business in the Yangtze River Delta to provide a new research perspective and policy ideas for the integrated development of the Yangtze River Delta.

This paper will try to measure the inter-provincial trade costs of the Yangtze River Delta based on the perspective of value-added trade to portray the actual situation and evolution trend of inter-provincial trade costs of the Yangtze River Delta under the division of labor in global value chains and analyze the promotion effect of inter-provincial trade costs of the Yangtze River Delta on inter-provincial trade development of the Yangtze River Delta accordingly, to provide a reference for the establishment of a unified market, industrial restructuring, transformation and upgrading of the Yangtze River Delta and the realization of high-quality integrated development goals. It also provides a reference for establishing a unified market, industrial restructuring, transformation and upgrading, and achieving high-quality integrated development. Secondly, the inter-provincial trade costs calculated from bilateral decomposition are used to reveal the status of industrial integration and trade patterns within the Yangtze River Delta to explore the inherent obstacles to deepen the optimization of industrial layout and value chain division of labor cooperation within the Yangtze River Delta.

## 2. Literature review

As an essential factor influencing trade development, trade costs are closely related to formulating and implementing many trade policy measures [14]. In the early stage of trade development, trade costs are mainly determined by transport costs, tariffs, and simple non-tariff barriers. They are measured using directly observable or quantifiable data, such as distance, transport, or tariff costs [15, 16]. The advantages of the direct measure are that it is simple and intuitive, and it is easy to know the level of trade costs based on the relevant metrics. However, with the development of trade, the types of non-tariff barriers are not only increasing but also more hidden; secondly, the components of trade costs are also increasing, and trade costs should include insurance costs, search and matching fees, international settlement costs, institutional and regulatory costs, and language communication costs, in addition to traditional trade costs such as transportation costs and tariff costs, to make the products reach the final consumers [17–19]. Again, it is difficult to collect data to measure traditional trade costs, and there are also statistical inconsistencies in the data of different countries and sectors. The existence of these problems makes the direct measurement method not well applied to the measurement of trade cost problems.

To effectively overcome the drawbacks of the direct measurement method, many scholars have started to adopt the indirect measurement method and have conducted multidimensional exploration for this purpose. Indirect measures are mainly divided into the price index method [20], the regression method [15, 21], and the inverse trade cost method using trade flows [14, 22, 23]. Initially, the gravity model in the regression method was more widely used [14]. However, the gravity model in the regression method is premised on the strict assumption of partial equilibrium between the two countries, which tends to omit some critical variables. For this reason, Anderson et al. introduced multilateral resistance factors for correction. They constructed a static multi-country general equilibrium model, which equipped the traditional gravity model with a micro basis and simplified the difficulty and method of measuring trade costs. On this basis, Novy introduced the iceberg-type bilateral trade costs and the symmetry assumption of successively eliminating trade costs so that the model application has a solid theoretical foundation. Since then, the principle of trade cost measurement based on the ex-post backward extrapolation of trade flows has gradually become the primary paradigm in this research field [14, 19, 24].

Regarding trade cost measurement research, domestic scholars have also made a lot of explorations and attempts [25–27]. Some scholars in China have also begun to apply the measurement method of foreign trade costs to the measurement research of inter-provincial trade costs in China. However, whether Sheng Bin et al. [28], Liu Jian [29] and Wang Fei et al. [30] use boundary effects or price indices to study the cost of China's interregional trade, Cui Xinsheng et al. [31] use institutional rule arrangements to facilitate exchange. Han Jiarong [32] and Li Ziruo et al. [33] used the Head Ries index to characterize administrative and geographical factors. In contrast, Zhao Jingmei et al. [34] used information search and logistics transportation costs to study the cost of interprovincial trade in China. They only analyzed some of the influencing factors of inter-regional trade costs. Xu Tongsheng et al. [26] are the first to apply the trade cost measure based on the ex-post backward projection of trade flows to the study of China's inter-provincial trade costs and measure China's inter-provincial trade costs accordingly and find that China's inter-provincial trade costs gradually decline. Significant industry differences exist, with the manufacturing industry having the lowest trade costs.

Pan Wenqing et al. [35] also measured China's inter-regional trade costs based on this principle and using China's regional input-output data from 1997–2007. Still, the results showed that China's inter-regional trade costs were generally rising, and the decline was only offered in the period after WTO accession. Since traditional trade statistics cannot exclude a large amount of double counting due to the trade of intermediate products under the value chain division of labor, the inter-provincial trade costs measured using the traditional statistical caliber trade data tend to distort the trade costs. They cannot effectively portray the actual situation of inter-provincial trade costs. Because of this, Yuan Kaihua et al. [36] attempted to extend the Novy model to the perspective of value-added trade. They tried to use China's regional input-output data to calculate and compare the cost of inter-regional trade. This extended method was later well used by scholars in the study of cost measurement of inter-provincial trade in China [37–40].

As can be seen above, the existing results on trade cost research mainly use traditional statistical caliber trade data for analysis, which cannot solve the problem of double counting in standard statistical caliber trade data under the division of labor in global value chains and thus can hardly reflect the actual situation of trade costs. Therefore, this paper will extend Novy's gravity model from the perspective of value-added trade to objectively measure the inter-provincial trade costs in the Yangtze River Delta to truly reflect the industrial division of labor and trade patterns in the Yangtze River Delta as a whole and between provinces, and to provide helpful policy insights for the establishment of a unified market in the Yangtze River Delta and the realization of high-quality integrated development goals. Compared with the existing literature, this paper is a pioneering study in the following aspects: first, by expanding the gravity model, the inter-provincial trade costs of the Yangtze River Delta are measured from the perspective of value-added, to objectively and genuinely reflect the inter-provincial trade costs of the Yangtze River Delta; second, the industry characteristics and division of labor pattern of inter-provincial trade costs of the Yangtze River Delta are dynamically analyzed; third, the data obtained by differential decomposition are used to reveal the role of inter-provincial trade cost factors in the development of inter-provincial trade in the Yangtze River Delta. The third is to use the data from the differential decomposition to reveal the contribution of inter-provincial trade cost factors in the development of inter-provincial trade in the Yangtze River Delta.

## 3. Rationale and data sources

### 3.1 Measurement of trade costs under value-added statistics

The main parameters and descriptions above are summarized in Table 1.

**Table 1. Parameters and descriptions.**

| Parameters | Descriptions | Notes |
|:---:|:---:|:---:|
| $x^{ij}$ | Nominal exports from country i to country j | $x^{ij} \geq 0$ |
| $x^{ji}$ | Nominal exports from country j to country i | $x^{ji} \geq 0$ |
| $x^{ii}$ | Domestic trade of country i | $x^{ii} \geq 0$ |
| $x^{jj}$ | Domestic trade of country j | $x^{jj} \geq 0$ |
| $y^i$ | Nominal income in country i | $y^i \geq 0$ |
| $y^j$ | Nominal income in country j | $y^j \geq 0$ |
| $y^w$ | World revenues | $y^w \geq 0; y^w = \Sigma_j y^j$ |
| $t^{ij}$ | Bilateral trade costs | $t^{ij} \geq 0$ |
| $t^{ii}$ | Domestic costs for country i | $t^{ii} \geq 0$ |
| $t^{jj}$ | Domestic costs for country j | $t^{jj} \geq 0$ |
| $\tau^{ij}$ | Bilateral trade costs | $\tau^{ij} \geq 0$ |
| $\sigma$ | Alternative resilience | $\sigma \geq 1$ |
| $\Pi^i$ | Price indices for country i | $\Pi^i \geq 0$ |
| $P^j$ | Price index for country j | $P^j \geq 0$ |

Anderson and Van [14] propose an indirect gravity model with micro-foundations for measuring trade costs based on the symmetry assumption.

$$x^{ij} = \frac{y^i y^j}{y^W} \left( \frac{t^{ij}}{\prod^i P^j} \right)^{1-\sigma} \tag{1}$$

However, the assumption in this Equation that the exogenous fixed bilateral trade costs($t^{ij}$) and the symmetry of bilateral trade costs ($t^{ij} = t^{ji}$) are not consistent with reality, and data on multilateral resistance price indices($\Pi^i$) and the Price index for country j ($P^j$)are challenging to obtain, making it difficult for the model to be widely applied. To overcome the shortcomings mentioned above of the model, Novy proposed an improved calculation model, pointing out that changes in bilateral trade costs can also affect the scale of domestic trade. For example, when the cost of employment between a country and other countries decreases, foreign trade exports will inevitably increase, which may crowd out a portion of domestic consumption and cause domestic trade to decline. From this, it can be concluded that:

$$x^{ii} = \frac{y^i y^i}{y^W} \left( \frac{t^{ii}}{\prod^i P^i} \right)^{1-\sigma} \tag{2}$$

According to Eq 2, the multilateral resistance factor can be calculated, and its expression is:

$$\prod^i P^i = \left( \frac{x^{ii}/y^i}{y^i/y^W} \right)^{\frac{1}{\sigma-1}} \times t^{ii} \tag{3}$$

From Eq 3, it can be inferred that the multilateral resistance factors of country i can be obtained by simply knowing the cost of domestic trade ($t^{ii}$). At this point, by using Eq 1, the gravity model of exports from country j to country i can be obtained, and it is multiplied by

both sides of Eq 1 to get a bidirectional gravity model equation. The specific expression is:

$$x^{ij}x^{ji} = \left(\frac{y^i y^j}{y^W}\right)^2 \left(\frac{t^{ij}t^{ji}}{\prod^i P^j \prod^j P^i}\right)^{1-\sigma} \tag{4}$$

By substituting the multilateral resistance factors ($\Pi^i P^i$ and $\Pi^j P^j$)obtained from Eq 3 into Eq 4, an asymmetric bilateral trade cost model equation relative to domestic trade costs can be obtained, specifically:

$$\left(\frac{t^{ij}t^{ji}}{t^{ii}t^{jj}}\right) = \left(\frac{x^{ij}/x^{ji}}{x^{ii}/x^{jj}}\right)^{\frac{1}{1-\sigma}} \tag{5}$$

By squaring both sides of Eq 5 and subtracting 1 at the same time, the expression for the equivalent trade cost of tariffs can be obtained:

$$\tau^{ij} = \left(\frac{t^{ij}t^{ji}}{t^{ii}t^{jj}}\right)^{\frac{1}{2}} - 1 = \left(\frac{x^{ii}x^{jj}}{x^{ij}x^{ji}}\right)^{\frac{1}{2(\sigma-1)}} - 1 \tag{6}$$

Eq 6, $\tau^{ij}$ represents the magnitude of bilateral trade costs $t^{ii}$ relative to domestic trade costs $t^{ii}$ and $t^{jj}$. $x^{ii}$ and $x^{jj}$ represents the domestic trade costs of countries i and j, respectively. $t^{ii}$ and $t^{jj}$ represent the domestic trade costs of countries i and j, respectively. The meanings of other variables are the same as those in Eq (1). This measurement method does not assume frictionless domestic trade and also reflects why the cost of bilateral trade is higher than that of domestic trade. From Eq 6, it can be seen that if bilateral trade flows $x^{ij}$ and $x^{ji}$ increases relative to domestic trade flows $x^{ii}$ and $x^{jj}$, bilateral trade becomes more straightforward compared to domestic trade; that is, bilateral trade costs $\tau^{ij}$ decrease, and vice versa. Therefore, this method can indirectly deduce the magnitude of bilateral trade costs from observable trade flows, becoming the central paradigm for indirectly measuring trade costs.

In the traditional inter-industry division of labor and trade model, the measure can provide an adequate standard of trade costs. However, with the in-depth development of the global value chain division of labor and trade, the rapid growth of business in intermediate products such as parts and components has become the main feature of international value chain trade, leading to the prevalence of third country (sectoral) effects and extensive double counting in bilateral trade statistics. To this end, this paper will introduce value-added trade data to improve and extend the above gravity model for measuring bilateral trade costs. This paper will illustrate the value-added accounting approach using a multi-country, multi-sector example and WWZ (2105). Based on the input-output relationship, it is obtained that:

$$X = AX + Y = (I - A)^{-1}Y = BY \tag{7}$$

In Eq 7, X represents each economy's total output matrix, Y represents each economy's final demand matrix, A represents the global production system matrix, I represents the identity matrix, and B represents the Leontief inverse matrix. It can be seen from this that a country's output comes from the final product demand of each economy. For illustration, the following will use three countries i, j, and k as examples to illustrate the value-added trade decomposition framework. Assuming $Va^i$ is the added value of country i, we can get Eq 8, "T" represents matrix transposition; "#" means matrix multiplication, which involves multiplying the corresponding elements in a matrix.

$$Va^i = (V^i)^T \# X^i \tag{8}$$

In Eq 8, V represents the vector of the department's added value coefficient (the ratio of the department's added value to total output), and X is the output matrix. According to the yield relationship caused by different demands, Eq 8 can be further expressed as:

$$X^i = \sum_j^3 \sum_k^3 B^{ik} Y^{kj} \tag{9}$$

In the Eq 9, B is the Leontief inverse matrix. Because the demand for a particular sector of a country's products includes both domestic and foreign parts, based on the different sources of final order and combined with Eq 8, Eq 7 can be further decomposed into:

$$Va^i = (V^i)^T \# \sum_j^3 B^{ij} Y^{ji} + (V^i)^T \# \sum_{j \neq i}^3 \sum_k^3 B^{ik} Y^{kj} = Va^{ii} + Va^{ij} \tag{10}$$

Eq 10, $Va^{ii}$ represents the added value caused by domestic final demand and $Va^{ij}$ the added value driven by foreign final demand. Eq 10 depicts the trade situation of intermediate component products on the global value chain. It can, therefore, be used to trace the actual sources of added value on the global value chain. Given this, the gravity model for measuring bilateral trade costs from the perspective of added value can be modified to:

$$\tau_h^\wedge ij = \left( \frac{Va_h^{ii} Va_h^{jj}}{Va_h^{ij} Va_h^{ji}} \right)^{\frac{1}{2(\sigma-1)}} - 1 \tag{11}$$

Based on the above principles, this paper will mainly use the value-added trade flows induced by the Yangtze River Delta and outer regions, as well as the inter-provincial demand of the Yangtze River Delta to make objective estimates of inter-provincial trade costs, and then portray the proper level and evolution of inter-provincial trade costs in the Yangtze River Delta.

## 3.2 Differential decomposition of inter-provincial trade flow promotion effects

Drawing on Xu Tongsheng [26], this paper extends a multi-country general equilibrium trade gravity model on trade costs developed by Anderson [14] to study the promotional effect of inter-provincial trade costs on development.

$$x^{ij} = \frac{y^i y^j}{y^c} \left( \frac{t^{ij}}{\prod^i P^j} \right)^{1-\sigma} \tag{12}$$

In Eq 12, the superscripts i, j, and c denote province i, province j, and China, respectively, $x^{ij}$ denotes exports from province i to province j, $y^i$ and $y^j$ denote the nominal income of provinces i and j, respectively, $y^c$ denotes the total size of the Chinese economy ($y^c = \sum_j y^j$), $t^{ij}$ denotes the cost of bilateral inter-provincial trade in the Yangtze River Delta, and $\sigma$ means the elasticity of substitution ($\sigma \geq 1$). $\Pi^i$ and $P^j$ denote the price indices of provinces i and j, respectively. By swapping the positions of the superscripts i and j in Eq (3) with each other, we can get the formula for measuring the size of exports from province j to province i.

$$x^{ji} = \frac{y^j y^i}{y^c} \left( \frac{t^{ji}}{\prod^j P^i} \right)^{1-\sigma} \tag{13}$$

Multiplying the left and right sides of Eq 12 and 13 gives:

$$x^{ij}x^{ji} = \left(\frac{y^i y^j}{y^c}\right)^2 \left(\frac{t^{ij}t^{ji}}{\prod^i P^j \prod^j P^i}\right)^{1-\sigma} \tag{14}$$

To facilitate the analysis of the impact of trade costs, economic growth, and multilateral resistance factors on the development of inter-provincial bilateral trade as measured by price indices, the logarithm of both sides of Eq 14 was continued to be taken simultaneously and then decomposed differentially. The specific results are as follows:

$$\Delta ln(x^{ij}x^{ji}) = 2\Delta ln\left(\frac{y^i y^j}{y^c}\right) + (1-\sigma)\Delta ln(t^{ij}t^{ji}) - (1-\sigma)\Delta ln(\Pi^i P^i \Pi^j P^j) \tag{15}$$

Dividing both sides of Eq 15 simultaneously by the first-order difference in bilateral trade flows $\Delta ln(x^{ij}x^{ji})$, we can obtain the following:

$$1 = \frac{2\Delta ln\left(\frac{y^i y^j}{y^c}\right)}{\Delta ln(x^{ij}x^{ji})} + \frac{(1-\sigma)\Delta ln(t^{ij}t^{ji})}{\Delta ln(x^{ij}x^{ji})} - \frac{(1-\sigma)\Delta ln(\Pi^i P^i \Pi^j P^j)}{\Delta ln(x^{ij}x^{ji})} \tag{16}$$

From Eq 16, the three items on the right-hand side indicate the contribution of economic growth, changes in inter-provincial bilateral trade costs, and multilateral resistance to the growth of inter-provincial bilateral trade flows, with economic growth, falling inter-provincial bilateral trade costs and rising multilateral opposition to foreign trade having a positive contribution to the development of inter-provincial bilateral trade, and the opposite contribution effect being negative.

Using the indirect measure of trade costs formula $\tau^{ij} = \left(\frac{t^{ij}t^{ji}}{t^{ii}t^{jj}}\right)^{\frac{1}{2}} - 1 = \left(\frac{x^{ji}x^{jj}}{x^{ij}x^{ji}}\right)^{\frac{1}{2(\sigma-1)}} - 1$, we can get the expression for the calculation of $t^{ij}t^{ji}$. We substitute it into Eq 16, and we can conclude that:

$$1 = \frac{2\Delta ln\left(\frac{y^i y^j}{y^c}\right)}{\Delta ln(x^{ij}x^{ji})} + \frac{2(1-\sigma)\Delta ln(1+\tau^{ij})}{\Delta ln(x^{ij}x^{ji})} - \frac{2(1-\sigma)\Delta ln(\varphi^i \varphi^j)}{\Delta ln(x^{ij}x^{ji})} \tag{17}$$

Given a large amount of double counting of trade flow data estimated by traditional statistical portals in the context of value chain division of labor, Eq 17 is further extended using the value-added trade flow estimation approach of Eq 16 to obtain a measure of the contribution effect of each factor to inter-provincial trade development as follows:

$$1 = \underbrace{\frac{2\Delta ln\left(\frac{y^i y^j}{y^c}\right)}{\Delta ln(Va^{ij}Va^{ji})}}\text{Economic growth affects} + \underbrace{\frac{2(1-\sigma)\Delta ln(1+\tau^{ij})}{\Delta ln(Va^{ij}Va^{ji})}}\text{Trade cost}$$

$$-\text{effectiveness} - \underbrace{\frac{2(1-\sigma)\Delta ln(\varphi^i \varphi^j)}{\Delta ln(Va^{ij}Va^{ji})}}\text{Multilateral resistance effect} \tag{18}$$

In Eq 18, the promotion effects of the first two items on the right side of the Equation on interprovincial trade development can be directly calculated from the relevant data, and the third multilateral resistance factor of foreign trade can be obtained through the value of the residual term.

### 3.3 Data sources

For the measurement of inter-provincial trade costs and value-added trade flows at the overall and inter-provincial levels of the Yangtze River Delta, this paper uses the 2012, 2015, and 2017 China Inter-Provincial and Inter-Regional Input-Output Tables. This paper also selects the GDP data published in the China Statistical Yearbook for 2012, 2015, and 2017 as the nominal income or economic size of the three provinces and one city in the Yangtze River Delta. For the value of the elasticity of substitution (usually 5–10), this paper still follows Novy and continues to set its value to 8.

## 4. Levels of inter-provincial trade costs in the Yangtze River Delta and their evolutionary trends

### 4.1 Overall inter-provincial trade costs of the Yangtze River Delta provinces/municipalities

According to Table 2, among the overall inter-provincial trade costs of the provinces/cities in the Yangtze River Delta measured under the value-added statistics, the lowest overall inter-provincial trade costs of the provinces/cities in the Yangtze River Delta in 2012, 2015, and 2017 are Shanghai, Anhui, and Jiangsu, whose trade costs are 19.19%, 16.83%, and 17.74%, respectively; while the highest overall inter-provincial trade costs of the provinces/municipalities in the Yangtze River Delta in the same period In the same period, the highest inter-provincial trade costs in the Yangtze River Delta provinces/cities are Zhejiang, Shanghai, and Anhui, with the corresponding trade costs of 22.84%, 21.54%, and 28.76% respectively. According to the trend of trade cost changes shown in the table, the inter-provincial trade cost in Shanghai has been increasing during the examination period, with an increase of 12.25% compared to 2012 and 13.26% compared to 2015. Inter-provincial trade costs in Jiangsu and Zhejiang provinces have decreased during the examination period. The decrease in inter-provincial trade costs in Jiangsu province was 6.46% and 7.24% in 2015 and 2017, and the reduction of inter-provincial trade costs in Zhejiang province was 9.60% and 13.40% in the same period. During the examination period, Anhui's inter-provincial trade costs, on the other hand, showed a trend of first will then rise, with a decrease of 24.98% in 2015 and a rise of 70.87% in 2017.

Compared with Xu Tongsheng et al. [26] findings, the overall inter-provincial trade cost levels of all provinces/municipalities in the Yangtze River Delta have declined substantially compared to the earlier period. This is mainly due to the deep integration of the Yangtze River Delta into the global value chain division of labor and the increasing degree of opening up to the outside world, forcing the domestic market to open up gradually as well, and the development of introduction up to the outside world has caused the domestic market integration to improve. From within the Yangtze River Delta, Shanghai's overall inter-provincial trade costs are on the rise, mainly because Shanghai has the fastest economic structural transformation and the highest proportion of service industries among the three provinces and one city in the Yangtze River Delta. Given the unique nature of service industries, its inter-provincial division

**Table 2. Overall inter-provincial trade costs of provinces/cities in the Yangtze River Delta under the value-added statistics.**

| Province/municipality | 2012 | 2015 | Change rate | 2017 | Change rate |
|---|---|---|---|---|---|
| Jiangsu | 20.45% | 19.13% | -6.46% | 17.74% | -7.24% |
| Shanghai | 19.19% | 21.54% | 12.25% | 24.40% | 13.26% |
| Zhejiang | 22.84% | 20.64% | -9.60% | 17.88% | -13.40% |
| Anhui | 22.44% | 16.83% | -24.98% | 28.76% | 70.87% |

of labor and cooperation is bound to lag behind manufacturing. All these negatively impact the value chain division of labor and cooperation between Shanghai and other provinces in the Yangtze River Delta, leading to a magnitude of sustained increase in trade costs. Shanghai, as the center of the economic development of the Yangtze River Delta, should play a more excellent leading and radiating role, significantly shifting the focus of cooperation from primary and secondary industries to tertiary industries to achieve the overall rapid development of the Yangtze River Delta regional economy.

## 4.2 Inter-provincial trade costs at the sectoral level of provinces/municipalities in the Yangtze River Delta

According to the results in Table 3, there are significant differences in inter-provincial trade costs at the sectoral level among provinces/municipalities in the Yangtze River Delta. In 2012, the sectors with the lowest inter-provincial trade costs at the sectoral level in Jiangsu, Shanghai, Zhejiang, and Anhui were S12, S12, S05, and S11, with trade costs of 3.51%, 1.68%, 2.03%, and 0.57%, respectively; the sectors with the highest inter-provincial trade costs are S28, with trade costs of 85.10%, 93.57%, 96.87%, and 87.59%, respectively. sectors were all S28, with trade costs of 85.10%, 93.57%, 96.87% and 87.59%, respectively. in 2017, the sectors with the lowest interprovincial trade costs in Jiangsu, Shanghai, Zhejiang, and Anhui became S11, S11, S07, and S02, with trade costs of 0.03%, 0.87%, 0.33%, and 1.73%, respectively; the highest trade sectors are all S35, with interprovincial trade costs of 131.75%, 116.55%, 102.27% and 119.15%, respectively. It can be seen that, on the one hand, the sectors with low trade costs are mostly manufacturing sectors, while the sectors with high trade costs are service sectors; on the other hand, there are large differences in inter-provincial trade costs at the sectoral levels of Jiangsu, Shanghai, Zhejiang, and Anhui, indicating that there is a significant complementary relationship of advantages at the sectoral levels of provinces/municipalities within the Yangtze River Delta. Provinces can vigorously develop their profitable industries, strengthen the division of labor and integration among sectoral levels in the region, improve the overall level of division of labor and cooperation, and strive to reduce inter-provincial trade costs to achieve high-level integrated development among sectors within the Yangtze River Delta region.

Although there is significant heterogeneity in inter-provincial trade costs at the sectoral level among provinces/municipalities in the Yangtze River Delta region, the inter-provincial trade costs of both sectors S11 and S12 in each province/municipality in the area are relatively low and belong to the dominant industrial industries in the Yangtze River Delta region. In contrast, the trade costs of both sectors, S28 and S35, are relatively high. This indicates that provinces/municipalities in the Yangtze River Delta have simultaneous competitive advantages or disadvantages in individual sectors. The provinces/municipalities in the Yangtze River Delta should focus on the sectors with higher trade costs, increase technological R&D and industrial upgrading, and focus on deepening the division of labor and integration with foreign or other domestic economic regions to realize the improvement of competitiveness and reduction of trade costs.

## 4.3 Inter-provincial bilateral trade costs in the Yangtze River Delta

From the results shown in Table 4, it is clear that the inter-provincial bilateral trade costs in the Yangtze River Delta fluctuated significantly during the examination period, and there are significant differences in the absolute level and fluctuation trend of inter-provincial bilateral trade costs. From the fluctuation trend, Shanghai-Suzhou bilateral trade costs have been slightly increasing, and bilateral trade costs rose from 38.15% in 2012 to 40.05% in 2015 and further rose to 41.12% in 2017; Zhejiang and Anhui, Suzhou and Shanghai and Anhui bilateral

**Table 3. Overall inter-provincial trade costs at the sectoral level in the YRD provinces under the value-added statistics.**

| Department | 2012 | | | | 2015 | | | | 2017 | | | |
|---|---|---|---|---|---|---|---|---|---|---|---|---|
| | Jiangsu | Shanghai | Zhejiang | Anhui | Jiangsu | Shanghai | Zhejiang | Anhui | Jiangsu | Shanghai | Zhejiang | Anhui |
| S01 | 14.73% | 15.96% | 14.35% | 17.99% | 11.99% | 15.57% | 14.48% | 10.76% | 11.14% | 26.00% | 12.11% | 18.70% |
| S02 | - | - | - | - | - | - | - | - | - | - | - | 1.73% |
| S03 | - | - | - | - | - | - | - | - | - | - | - | - |
| S04 | - | - | - | - | - | - | - | - | - | - | - | - |
| S05 | - | - | 2.03% | - | - | - | 5.39% | - | - | - | 1.85% | 10.84% |
| S06 | 12.85% | 20.05% | 17.11% | 23.16% | 10.77% | 20.13% | 13.87% | 16.98% | 16.91% | 33.03% | 15.45% | 26.08% |
| S07 | 13.79% | 11.52% | 12.10% | 15.56% | 11.03% | 11.94% | 7.57% | 8.80% | 16.90% | 13.38% | 0.33% | 14.57% |
| S08 | 32.85% | 30.14% | 31.31% | 33.07% | 27.54% | 26.75% | 23.06% | 24.58% | 38.24% | 38.98% | 27.68% | 47.73% |
| S09 | 12.85% | 12.30% | 13.78% | 13.75% | 10.68% | 14.50% | 17.91% | 7.41% | 14.62% | 26.34% | 10.99% | 29.59% |
| S10 | 9.27% | 5.37% | 7.29% | 7.46% | 8.32% | 8.55% | 3.71% | 1.45% | 4.32% | 7.56% | 1.21% | 9.05% |
| S11 | 5.32% | - | - | 0.57% | 3.02% | - | - | - | 0.03% | 0.87% | - | 7.43% |
| S12 | 3.51% | 1.68% | 2.44% | 6.34% | 2.71% | - | 1.08% | 0.41% | 0.15% | 2.97% | - | 9.98% |
| S13 | 9.22% | 12.35% | 12.15% | 11.52% | 7.77% | 13.26% | 17.50% | 5.87% | 1.46% | 10.50% | - | 18.69% |
| S14 | 6.70% | 5.93% | 3.62% | 1.38% | 3.17% | 10.65% | 8.17% | - | 2.50% | - | 5.47% | 10.22% |
| S15 | 19.92% | 14.51% | 15.61% | 16.54% | 14.20% | 15.63% | 16.68% | 10.75% | 14.11% | 12.08% | 10.46% | 17.02% |
| S16 | 20.27% | 22.44% | 26.17% | 24.15% | 19.97% | 21.38% | 18.90% | 18.30% | 27.36% | 29.82% | 21.86% | 20.18% |
| S17 | 31.57% | 32.19% | 31.17% | 33.89% | 25.81% | 36.97% | 19.99% | 25.52% | 41.13% | 54.52% | 32.83% | 36.75% |
| S18 | 35.50% | 29.35% | 33.61% | 39.22% | 29.76% | 23.22% | 25.52% | 31.12% | 20.57% | 28.40% | 19.73% | 31.83% |
| S19 | 22.43% | 16.93% | 22.45% | 27.33% | 16.21% | 16.16% | 17.02% | 16.90% | 23.69% | 22.39% | 17.74% | 22.76% |
| S20 | 16.01% | 8.92% | 22.01% | 22.49% | 14.79% | 14.67% | 13.49% | 13.49% | 17.29% | 23.72% | 8.75% | 15.41% |
| S21 | 23.46% | 19.35% | 20.17% | 26.02% | 16.37% | 13.00% | 15.69% | 15.65% | 20.29% | 21.13% | 3.39% | 11.09% |
| S22 | 9.38% | 3.28% | 7.52% | 16.73% | 10.46% | 2.60% | 3.65% | 9.96% | - | - | - | 3.30% |
| S23 | - | - | - | - | -7.18% | 1.02% | - | - | - | - | - | - |
| S24 | - | - | - | - | 0.25% | - | - | - | 3.91% | 9.71% | - | 10.64% |
| S25 | 7.68% | 6.30% | 6.58% | 4.28% | 5.40% | 5.22% | 1.17% | 1.49% | 3.36% | 8.36% | 6.33% | 9.24% |
| S26 | 6.68% | 11.05% | 9.97% | 7.30% | 4.54% | 11.38% | 7.06% | 3.59% | - | 14.56% | 8.69% | 4.10% |
| S27 | 9.26% | 8.00% | 8.94% | 7.61% | 5.66% | 8.92% | 12.48% | - | 43.20% | 101.16% | 91.50% | 108.13% |
| S28 | 85.10% | 93.57% | 96.87% | 87.59% | 77.21% | 86.50% | 81.81% | 77.48% | 10.45% | 17.93% | 8.33% | 21.31% |
| S29 | 11.78% | 10.54% | 10.38% | 5.91% | 6.57% | 12.71% | 8.38% | 2.01% | 2.35% | 5.10% | 2.34% | 19.15% |
| S30 | 5.04% | - | 7.08% | 3.17% | 3.77% | 1.89% | 7.18% | - | 8.88% | 23.27% | 14.97% | 19.55% |
| S31 | 15.65% | 11.70% | 11.54% | 15.28% | 10.87% | 13.95% | 7.00% | 7.47% | 35.49% | 26.40% | 28.67% | 32.73% |
| S32 | 17.13% | 10.79% | 19.07% | 19.02% | 14.60% | 12.01% | 15.07% | 14.81% | 4.02% | 11.56% | 5.01% | 14.54% |
| S33 | 10.17% | 4.99% | 12.57% | 7.25% | 6.63% | 7.22% | 5.86% | 0.12% | 14.14% | 17.56% | 12.40% | 30.15% |
| S34 | 33.20% | 12.72% | 37.62% | 31.59% | 23.98% | 12.76% | 22.93% | 12.63% | 2.93% | 9.42% | - | 14.47% |
| S35 | 8.98% | 5.08% | 11.44% | 2.62% | 6.36% | 5.91% | 4.73% | - | 131.75% | 116.55% | 102.27% | 119.15% |
| S36 | 23.28% | 24.90% | 25.36% | 28.18% | 21.45% | 25.58% | 30.44% | 18.65% | 8.47% | 11.39% | 27.34% | 26.71% |
| S37 | 34.82% | 38.31% | 32.31% | 38.76% | 32.55% | 41.86% | 32.03% | 34.27% | 44.33% | 56.78% | 40.05% | 41.14% |
| S38 | 14.31% | 8.85% | 15.92% | 11.54% | 13.47% | 8.36% | 14.48% | 7.02% | 9.44% | 5.12% | 10.57% | 13.39% |
| S39 | 38.04% | 28.46% | 40.52% | 33.61% | 34.43% | 27.78% | 34.80% | 27.50% | 42.75% | 45.65% | 33.02% | 42.46% |
| S40 | 65.39% | 69.07% | 52.33% | 75.39% | 62.43% | 64.89% | 28.72% | 63.05% | 104.75% | 103.95% | 91.92% | 116.46% |
| S41 | 22.46% | 19.94% | 20.62% | 22.37% | 19.25% | 32.20% | 20.42% | 17.35% | 33.99% | 37.49% | 30.99% | 37.99% |
| S42 | 39.12% | 23.39% | 40.17% | 54.36% | 37.57% | 26.13% | 40.27% | 28.09% | 21.69% | 32.40% | 20.92% | 70.89% |

Note: When the trade flow is 0, the overall inter-provincial trade cost at the sectoral level of the Yangtze River Delta provinces under the value-added statistics is not calculable, or data is missing and is indicated by "-."

**Table 4. Inter-provincial bilateral trade costs in the Yangtze River Delta under the value-added statistics.**

| Province/municipality | 2012 | | | | 2015 | | | | 2017 | | | |
|---|---|---|---|---|---|---|---|---|---|---|---|---|
| | Jiangsu | Shanghai | Zhejiang | Anhui | Jiangsu | Shanghai | Zhejiang | Anhui | Jiangsu | Shanghai | Zhejiang | Anhui |
| Jiangsu | - | 38.15% | 44.74% | 41.15% | - | 40.05% | 39.18% | 32.08% | - | 41.12% | 30.48% | 33.97% |
| Shanghai | 38.15% | - | 40.51% | 34.57% | 40.05% | - | 37.45% | 31.13% | 41.12% | - | 30.38% | 48.24% |
| Zhejiang | 44.74% | 40.51% | - | 41.78% | 39.18% | 37.45% | - | 31.79% | 30.48% | 30.38% | - | 42.68% |
| Anhui | 41.15% | 34.57% | 41.78% | - | 32.08% | 31.13% | 31.79% | - | 33.97% | 48.24% | 42.68% | - |

trade costs all showed a fluctuation trend of first decreasing and then increasing, and bilateral trade costs respectively fell from 41.78% in 2012, 41.15% and 34.57% in 2012 to 31.79%, 32.08% and 31.13% in 2015, and then rose to 42.68%, 33.97% and 48.24% in 2017; bilateral trade costs of Jiangsu, Zhejiang, and Shanghai-Zhejiang have been showing a decreasing trend, with bilateral trade costs falling from 44.74% and 40.51% in 2012 to 39.18% and 37.45% in 2015, and further decreased to 30.48% and 30.38% respectively in 2017.

In terms of absolute level, in 2012, the bilateral trade cost of Shanghai and Anhui was the lowest, with a trade cost of 34.57%, followed by the bilateral trade cost of Shanghai and Suzhou, with a trade cost of 38.15%; the highest trade cost was Zhejiang and Suzhou, with a high of 44.74%. Compared with 2012, the bilateral trade cost of Shanghai and Anhui is still the lowest in 2015, at 31.13%; the provinces with the highest bilateral trade cost have changed, with the most increased bilateral trade cost of Shanghai and Suzhou, at 40.05%. But in 2017, the bilateral trade cost of Shanghai and Zhejiang became the lowest, only 30.38%, while the bilateral trade cost level of Shanghai and Anhui was the highest, reaching 48.24%. It can be seen that not only the level of economic development, industrial structure, and regional complementarity are essential factors influencing the change of trade costs and the formation of the division of labor pattern, but also the economic distance caused by geographical location has an essential impact on trade costs. Compared with other provinces, the distance between Shanghai and Anhui is the farthest, which becomes a necessary obstacle to decreasing bilateral trade costs between Shanghai and Anhui.

## 5. Promotional effects of inter-provincial trade costs on inter-provincial trade development in the Yangtze River Delta

### 5.1 Promotional effect of inter-provincial trade cost on the overall inter-provincial trade development of the provinces/municipalities in the Yangtze River Delta

It is easy to know from Eq 11 that trade flow is negatively related to the trade cost factor and positively associated with economic growth and multilateral resistance factors. This implies that financial increase, multilateral resistance rise, and trade cost decrease can effectively promote the development of inter-provincial trade in the Yangtze River Delta.

**Table 5. Decomposition of the effects of overall inter-provincial trade development in the Yangtze River Delta provinces/municipalities, 2012–2017.**

| Region | The growth rate of inter-provincial value-added trade flows | Contribution to income growth | Contribution to declining trade costs | Contribution of declining multilateral resistance |
|---|---|---|---|---|
| Jiangsu | -13.25% | 80.22% | 27.95% | 8.17% |
| Shanghai | 27.15% | 80.91% | -55.23% | -74.32% |
| Zhejiang | -21.72% | 53.98% | 35.99% | -10.03% |
| Anhui | 28.16% | 112.56% | -83.25% | -70.69% |

According to Table 5, the growth rates of value-added trade export flows of each of Jiangsu, Shanghai, Zhejiang, and Anhui to other provinces/municipalities within the Yangtze River Delta during 2012–2017 are -13.25%, 27.15%, -21.72% and 28.16% in that order. In terms of the factors influencing the change in value-added trade flows, Jiangsu and Zhejiang have negative growth in value-added export trade flows, and the main controlling elements for the change are economic growth and declining trade costs. For Jiangsu, the contribution of economic growth is the largest, reaching 80.22; the assistance of the decline in trade costs is the second largest, at 27.95%; the contribution of multilateral resistance to corrosion is the smallest, at 8.17%. For Zhejiang, where trade flows are also negative, the contribution of economic growth is 53.98%, the donation of falling trade costs is 35.99%, and the contribution of decreasing multilateral resistance is -10.03%.

For Shanghai and Anhui, the value-added export trade flows grew positively during the period under examination, with growth rates of 27.15% and 28.16%, respectively. Economic growth is also the main influencing factor for the development of inter-provincial trade in Shanghai and Anhui, with contribution rates of 80.91% and 112.56%, respectively; the contribution of the decline in multilateral resistance has the second highest impact on the trade flows in Shanghai (-74.32%), and the contribution of the reduction in trade costs has the second highest impact on the trade flows in Anhui (-83.25%); the influencing factors that contribute the least to the changes in trade flows in Shanghai and Anhui The factors contributing least to the changes in trade flows in Shanghai and Anhui are the decline in trade costs and the reduction in multilateral resistance, respectively, with their contributions of -55.32% and -70.69%, in that order.

## 5.2 Contribution effect of inter-provincial trade costs on inter-provincial bilateral trade development

To further understand the regional differences of inter-provincial trade development and its influencing factors among provinces/municipalities in the Yangtze River Delta, a differential decomposition of the growth of inter-provincial bilateral value-added trade flows in the Yangtze River Delta and the contribution rate of its related factors is made. The specific results are shown in Table 6. It is the largest, up to 335%. Suzhou-Shanghai and Anhui-Shanghai's bilateral value-added trade flows declined significantly with -43.51% and -45.65%, respectively. From the results of differential decomposition, there are significant differences in the factors that contribute most to the changes in bilateral value-added trade flows among provinces, among which economic growth factors contribute most to the changes of bilateral value-added trade flows of Shanghai-Suzhou, Shanghai-Anhui, Suzhou-Shanghai, Suzhou-Anhui, Zhejiang-Anhui, Anhui-Shanghai, Anhui-Suzhou, and Anhui-Zhejiang, with their corresponding contribution rates of 160.95%, 632.98%, 160.95%, 87.24%, 132.04%, 632.04%, and 632.65%, respectively. The corresponding contribution rates are 160.95%, 632.98%, 160.95%, 87.24%, 132.04%, 632.98%, 87.24%, and 132.04%, respectively; the factor of decreasing trade cost has the most significant contribution to the change of bilateral value-added trade flows of Shanghai, Zhejiang, Jiangsu, Zhejiang, Shanghai, and Zhejiang-Suzhou, with the corresponding contribution rates of 45.18%, 91.45%, 45.18%, and 91.45% respectively; the factor of decreasing multilateral resistance has the most significant contribution to the change of bilateral value-added trade flows of Shanghai, Zhejiang, Shanghai, Anhui, Shanghai, and Zhejiang. Trade flows also have a specific contribution to the growth of trade flows, whose contribution rates are -17.73%, -399.60%, -17.73%, and -399.60%, respectively. In addition, as the bilateral trade costs of Shanghai-Suzhou, Shanghai-Anhui, Suzhou-Shanghai, Zhejiang-Anhui, Anhui-Shanghai, and Anhui-Zhejiang have increased instead of decreased during the period under

**Table 6. Effect decomposition of inter-provincial bilateral trade development in the Yangtze River Delta, 2012–2017.**

| Region | Inter-provincial trade growth rate | The contribution rate of income growth | The contribution rate of declining trade costs | The contribution rate of declining multilateral resistance |
|---|---|---|---|---|
| Shanghai-Jiangsu | 207.68% | 160.95% | -53.87% | 7.08% |
| Shanghai-Zhejiang | 335.00% | 37.09% | 45.18% | -17.73% |
| Shanghai-Anhui | 112.76% | 632.98% | -932.58% | -399.60% |
| Jiangsu-Shanghai | -43.51% | 160.95% | -53.87% | 7.08% |
| Jiangsu-Zhejiang | 141.30% | 55.41% | 91.45% | 46.86% |
| Jiangsu- Anhui | 49.21% | 87.24% | 66.67% | 53.91% |
| Zhejiang-Shanghai | 133.55% | 37.09% | 45.18% | -17.73% |
| Zhejiang-Jiangsu | 102.78% | 55.41% | 91.45% | 46.86% |
| Zhejiang- Anhui | 41.93% | 132.04% | -12.72% | 19.32% |
| Anhui-Shanghai | -45.65% | 632.98% | -932.58% | -399.60% |
| Anhui-Jiangsu | 100.57% | 87.24% | 66.67% | 53.91% |
| Anhui-Zhejiang | 41.42% | 132.04% | -12.72% | 19.32% |

examination, resulting in the inter-provincial trade cost factor becoming an obstacle to the development of bilateral trade.

The promotion effect of interprovincial trade development and its influencing factors in the Yangtze River Delta shows that the economic growth factor makes the most significant contribution to interprovincial trade development, and this result is consistent with the leading research findings at home and abroad. The gift of inter-provincial trade cost factors to inter-provincial trade development in the Yangtze River Delta is much lower than that of economic growth factors, which is mainly due to the minimal decrease in inter-provincial trade costs in the Yangtze River Delta and the fact that some of them have increased instead of decreased, which limits the promotion effect of trade cost factors on trade development.

## 6. Conclusion and policy suggestions

Given that the value-added statistical port data can solve the problem of duplicate statistics caused by a large amount of intermediate product trade in the global value chain division of labor to effectively reflect the actual situation of the worldwide value chain division of labor. To this end, this paper uses value-added statistics port trade data to objectively measure and analyze inter-provincial trade costs in the Yangtze River Delta. It uses the differential decomposition method to explore the contribution effect of inter-provincial trade costs to the growth of inter-provincial trade flows in the Yangtze River Delta. The study draws the following conclusions:

(1) Significant differences exist in the fluctuation trends of the overall inter-provincial trade costs of the provinces/municipalities in the Yangtze River Delta. During the examination period, the overall inter-provincial trade costs of Jiangsu and Zhejiang have been on a downward trend, with a decline of 2.71% and 4.96%; the overall inter-provincial trade costs of Anhui are on a trend of first going up and then rising, with an overall increase of 6.32%; the overall inter-provincial trade costs of Shanghai have been on an upward trend, with a rise of 5.21% in trade costs. Due to the rapid transformation of Shanghai's industrial structure, the

proportion of the service industry is significantly higher than that of the other three provinces in the region. The service industry has many labor-intensive sectors, which can absorb more labor and effectively solve local employment problems. However, the development of the service industry lags the manufacturing industry, making it difficult for many services to operate on a large scale to achieve cost reduction and efficiency increase.

Moreover, many services as products are not traceable due to high transaction costs. This inevitably leads to an increase in the inter-provincial trade costs in Shanghai as the proportion of the service industry in the total economy increases. In addition, the division of labor cooperation in the service industry started relatively late, especially the inter-provincial division of labor cooperation, which lags far behind the manufacturing industry, inevitably leading to a small flow of inter-provincial trade in services and a large flow of intra-provincial business. Based on the reverse inference of trade flow, the cost of inter-provincial work in Shanghai will naturally be relatively high.

(2) Difference in overall inter-provincial trade costs at the departmental level in the Yangtze River Delta mainly manifests in the service sector's relatively high trade costs. The difference in inter-provincial trade costs among various provinces/cities in the Yangtze River Delta is primarily reflected in the overall higher inter-provincial trade costs of the service sector compared to the trade costs of the primary and secondary industry sectors. This is in line with the pace of industrial structure transformation and the characteristics of low tradability and high transaction costs of services. After the development of the first and second industries reaches a particular stage, they will gradually transition to the third industry, and the proportion of the service industry will increase. This inevitably leads to a weaker breadth and depth of inter-provincial division of labor cooperation in the service sector compared to the primary and secondary industry sectors, and the inter-provincial trade costs in the service sector will naturally be relatively high.

(3) Differences in inter-provincial trade costs in the Yangtze River Delta are also reflected in regional differences, namely significant differences in inter-provincial bilateral trade costs. At the beginning of the inspection period, the cost of bilateral trade between Shanghai and Anhui was the lowest, while the price between Zhejiang and Jiangsu was the highest. However, at the end of the inspection period, the cost of bilateral trade between Shanghai and Anhui was the highest, while the cost of bilateral trade between Shanghai and Zhejiang was the lowest. With the development and expansion of inter-provincial business in the Yangtze River Delta, the impact of economic distance factors on trade costs is becoming increasingly prominent. Therefore, it is crucial to strengthen infrastructure construction, especially the construction of transportation infrastructure, reduce the logistics costs of trade targets, minimize the negative impact of economic distance factors on the decline of interprovincial trade costs, and thus promote the growth and integrated development of interprovincial trade in the Yangtze River Delta.

(4) Differential decomposition method was used to decompose the promoting effect of inter-provincial trade cost factors on the development of inter-provincial trade in the Yangtze River Delta. It was found that the contribution rate of inter-provincial trade cost factors to the development of inter-provincial business needs to be further improved. The contribution rate of economic growth factors to the development of inter-provincial trade is much higher than that of trade cost factors. On the one hand, this is because the Yangtze River Delta has a high degree of economic openness and dependence on foreign trade. The development of foreign exchange has a particular substitution effect on the development of inter-provincial business in the region, which limits the growth of inter-provincial work in the Yangtze River Delta.

On the other hand, the cost of bilateral trade between some provinces in the Yangtze River Delta is indeed high, and the decline is relatively slow. Some provincial trade costs have also

increased to a certain extent, seriously hindering the promotion effect of the decrease in regional trade costs on the growth of restricted trade. Therefore, it is necessary to vigorously promote the rapid development of inter-provincial business in the Yangtze River Delta by reducing the cost of inter-provincial work and improving the integrated development level of the Yangtze River Delta.

Based on the effective measurement of inter-provincial trade costs in the Yangtze River Delta using trade data of value-added statistics, this paper further analyzes the promotion effect of inter-provincial trade costs on inter-provincial trade development in the Yangtze River Delta using the difference decomposition method. It provides a new policy design idea and path reference for deepening the division of labor and cooperation in the value chain within the Yangtze River Delta and achieving a higher level of integrated development. First, the three provinces and one city in the Yangtze River Delta should give full play to the advantages of their respective industrial sectors based on the principle of complementarity, deepen the inter-provincial sectoral division of labor and cooperation, cut inter-provincial trade costs, and promote the development of inter-provincial trade. Second, strengthen the division of labor and cooperation between Shanghai and the other three provinces, reduce the overall inter-provincial trade costs of Shanghai, give full play to the central role of Shanghai in the economic development of the Yangtze River Delta, enhance the radiation and driving effect of Shanghai's development on other three provinces, and promote the integrated development of the Yangtze River Delta. Finally, to address the problems of high inter-provincial trade costs in the service sector and lagging inter-provincial division of labor and cooperation in the service sector, provinces/municipalities in the Yangtze River Delta, especially Shanghai, should use digital technology and other means to accelerate digital transformation and innovation in the service sector according to the unique attributes of the service sector, to facilitate the inter-provincial division of labor and cooperation in the service sector, change the situation of generally high inter-provincial trade costs in the service sector and enhance the role of declining inter-provincial trade costs in promoting inter-provincial trade growth in the Yangtze River Delta.

1. This article only provides a preliminary study on the cost measurement and promotion effect decomposition of inter-provincial trade. As for how to theoretically demonstrate the rationality of the cost mechanism of inter-provincial business and how to explain its measurement and promotion effect results, further in-depth research is needed.

2. Due to the small input-output data of some departments in the "China Province City Interregional input-output Table," rounding and other methods used in the calculation process of this article may inevitably result in errors.

3. This article mainly examines the contribution rate of income growth and declining trade costs to the development of inter-provincial trade. Still, it fails to systematically incorporate other influencing factors into the research scope.

4. This article only proposes some preliminary suggestions and ideas from the perspective of reducing inter-provincial trade costs based on research results. It fails to design more systematic and targeted measures from a strategic perspective to address the situation of local protection and market segmentation and establish a unified regional market.

## Author Contributions

**Data curation:** Renxiang Lu.

**Formal analysis:** Yang Wang.

**Funding acquisition:** Renxiang Lu.

**Resources:** Yang Wang, Huanyue Tong.

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
