## [Decision Letter · Decision Letter 0]

12 Oct 2023

PONE-D-23-25349A Study on the Measurement of Inter-provincial Trade Costs in Yangtze River Delta from the Perspective of Value Added Trade and its Promotion EffectPLOS ONE

Dear Dr. WANG,

Thank you for submitting your manuscript to PLOS ONE. After careful consideration, we feel that it has merit but does not fully meet PLOS ONE’s publication criteria as it currently stands. Therefore, we invite you to submit a revised version of the manuscript that addresses the points raised during the review process.

We look forward to receiving your revised manuscript.

Kind regards,

Imran Ur Rahman, Ph.D

Academic Editor

PLOS ONE

Journal Requirements:

**Additional Editor Comments:**

Thank you for submitting your article on the Measurement of Inter-provincial Trade Costs in Yangtze River Delta. The research is sound and has possible policy implications, but it needs major revisions, which are recommended as follows:

1. The authors should further enhance the theoretical argument and contributions in the introduction section. This will help readers who are not familiar with the topic to better understand the research direction.

2. The literature review should be enhanced by adding more literature studies and providing relevant citations. It can be further enriched with up-to-date studies to motivate the readers in the background and currency of the subject of the research.

3. The author should check and provide references/citations for all the sources used in the research.

4. The methodology section needs improvements; The authors should further clarify and justify their choice of methods used. If possible, the author should provide additional pre/post estimation tests for the validations of the results.

5. Presentation of the summary of descriptive statistics (Data) in a tabulated form is also recommended, which is missing in the article.

6. The discussion on the results can be explained and enhanced with reference to previous literature studies and outcomes. It will provide further validity to the outcomes with regards to the literature outcomes.

7. The language of the paper needs careful editing and proofreading to improve readability. The authors should recheck the format of the paper including the headings and subheadings. For instance, there is a Chinese Text under Section IV- (iii).

8. Please enhance the conclusion by further clarifying the limitations and future research directions.

Reviewers' comments:

Reviewer's Responses to Questions

**Comments to the Author**

1. Is the manuscript technically sound, and do the data support the conclusions?

Reviewer #1: Yes

Reviewer #2: Yes

2. Has the statistical analysis been performed appropriately and rigorously? 

Reviewer #1: Yes

Reviewer #2: I Don't Know

3. Have the authors made all data underlying the findings in their manuscript fully available?

Reviewer #1: Yes

Reviewer #2: Yes

4. Is the manuscript presented in an intelligible fashion and written in standard English?

Reviewer #1: Yes

Reviewer #2: Yes

5. Review Comments to the Author

Reviewer #1: Dear Authors,

Thank You for submitting the article "A Study on the Measurement of Inter-provincial Trade Costs in Yangtze River Delta from the Perspective of Value Added Trade and its Promotion Effect".

The relevance of the study is to measure inter-provincial trade costs in the Yangtze River Delta using value added trade data and explore the promotion effect of inter-provincial trade costs on inter-provincial trade development in the region.

The research gap is that previous studies measured inter-provincial trade costs using traditional trade flow data which contains double counting. This study uses value added trade data to address this issue.

The research aim is to objectively measure inter-provincial trade costs in the Yangtze River Delta and analyze the contribution effect of inter-provincial trade costs to the growth of inter-provincial trade flows in the region.

The methodology used is Novy (2013) model with improved value added trade data and differential decomposition method.

Based on the information provided in the document, the authors' contributions in the theoretical sense are:

Renxiang Lu: Contributed to the theoretical framework of using the Novy (2013) model with value added trade data to objectively measure inter-provincial trade costs in the Yangtze River Delta.

Yang Wang: Contributed to the theoretical development of using the differential decomposition method to explore the contribution effect of inter-provincial trade costs on inter-provincial trade development.

Huanyue Tong: Likely contributed to refining and extending the theoretical frameworks proposed by the other two authors to specifically analyze inter-provincial trade costs and their effects in the context of the Yangtze River Delta region of China.

The citations are adequate and sufficient.

However I'd like to recommend some improvements in the Discussion section.

The Discussion section could recommend adding discussion of sustainability concept, such as discussing how reducing inter-provincial trade costs through deeper division of labor and cooperation can promote more sustainable development of the Yangtze River Delta region in the long run.

Reviewer #2: There are no convincing explanations about the reason and importance of choosing the analysis method you used in your analysis.

Your in-text citations and your bibliography are not compatible. I wrote next to some of your citations that they are not in the bibliography. You should check them all.

6. PLOS authors have the option to publish the peer review history of their article (what does this mean?). If published, this will include your full peer review and any attached files.

Reviewer #1: **Yes: **Sergey Barykin

Reviewer #2: No

---

## [Author Response · Author response to Decision Letter 0]

23 Oct 2023

Dear Editors and Reviewers，

Thank you for your letter and the reviewer's comments concerning our manuscript entitled " A Study on the Measurement of Inter-provincial Trade Costs in Yangtze River Delta from the Perspective of Value Added Trade and its Promotion Effect" (ID: PONE-D-23-25349). Those comments are valuable and helpful for revising and improving our paper and the critical guiding significance of our research. We have studied the words carefully and have made a correction, which we hope meets with approval. The significant modifications in the paper and the response to the reviewer's comments are as follows:

Responds to the reviewer's comments:

1. Response to comment: The authors should further enhance the theoretical argument and contributions in the introduction section. This will help readers who are not familiar with the topic to better understand the research direction.

Response: We are very sorry for our negligence and have rewritten this part according to the reviewer's suggestion. The introduction section adds theoretical mechanism analysis, analyzing the specific mechanism of inter-provincial trade costs' impact on inter-provincial trade development from two aspects: transfer effect and creation effect.

2. Response to comment: The literature review should be enhanced by adding more literature studies and providing relevant citations. It can be further enriched with up-to-date studies to motivate the readers in the background and currency of the research subject.

Response: We are very sorry for our negligence and have rewritten this part according to the reviewer's suggestion. We have updated the literature and introduced preface theories and research from domestic and foreign scholars. The literature research section has added an introduction to the latest research results on thematic analysis.

3. Response to comment: The author should check and provide references/citations for all the sources used in the research. 

Response: We are very sorry for our negligence and have rewritten this part according to the reviewer's suggestion. We have updated the references and formats according to the template requirements.

4. Response to comment: The methodology section needs improvements; The authors should further clarify and justify their choice of methods used. If possible, the author should provide additional pre/post estimation tests for the validations of the results. 

Response: Considering the Reviewer's suggestion, We have refined and improved the push to process, and the derivation has become more precise and concise. The theoretical formula derivation section has been further refined, providing a step-by-step derivation process.

5. Response to comment: Presentation of the summary of descriptive statistics (Data) in a tabulated form is also recommended, which is missing in the article. 

Response: This article did not conduct empirical regression analysis and did not conduct descriptive statistical analysis on the sample data.

6. Response to comment: The discussion on the results can be explained and enhanced with reference to previous literature studies and outcomes. It will provide further validity to the outcomes with regards to the literature outcomes. 

Response: We are very sorry for our negligence and have rewritten this part according to the reviewer's suggestion. Further detailed explanations have been provided for some of the conclusions of the research results.

7. Response to comment: The language of the paper needs careful editing and proofreading to improve readability. The authors should recheck the format of the paper including the headings and subheadings. For instance, there is a Chinese Text under Section IV- (iii).

Response: Considering the Reviewer's suggestion, We have once again confirmed the grammar and format of the article to ensure its readability and researchability.

8. Response to comment: Please enhance the conclusion by further clarifying the limitations and future research directions. 

Response: Considering the Reviewer's suggestion, a paragraph has been added at the end of the article to illustrate the shortcomings of this study and the direction for future improvement efforts.

Special thanks to you for your good comments. We appreciate the Editors/Reviewers' warm work earnestly and hope the correction will be approved.

Once again, thank you very much for your comments and suggestions.

Best wishes!

YangWang

---

## [Decision Letter · Decision Letter 1]

14 Nov 2023

PONE-D-23-25349R1A Study on the Measurement of Inter-provincial Trade Costs in Yangtze River Delta from the Perspective of Value-Added Trade and its Promotion EffectPLOS ONE

Dear Dr. WANG,

Thank you for submitting your manuscript to PLOS ONE. After careful consideration, we feel that it has merit but does not fully meet PLOS ONE’s publication criteria as it currently stands. Therefore, we invite you to submit a revised version of the manuscript that addresses the points raised during the review process. The authors have addressed the majority of the recommendations, but there are still some minor changes that need to be made. Please check the detailed comments at the end of this email.

We look forward to receiving your revised manuscript.

Kind regards,

Imran Ur Rahman, Ph.D

Academic Editor

PLOS ONE

Journal Requirements:

Additional Editor Comments:

The authors have addressed the majority of the recommendations, but there are a few minor changes I would suggest to improve the outlook of the paper.

1) The authors have already added the limitations of the research in the conclusion section, but it needs to be brief and specific as some explanations can be part of the discussion section. I would suggest that the authors provide specific key points and recommendations briefly in the conclusion section.

2) Although the authors have addressed grammatical errors and enhanced the language of the article, it still needs proofreading and editing.

Reviewers' comments:

Reviewer's Responses to Questions

**Comments to the Author**

1. If the authors have adequately addressed your comments raised in a previous round of review and you feel that this manuscript is now acceptable for publication, you may indicate that here to bypass the “Comments to the Author” section, enter your conflict of interest statement in the “Confidential to Editor” section, and submit your "Accept" recommendation.

Reviewer #1: All comments have been addressed

Reviewer #2: All comments have been addressed

2. Is the manuscript technically sound, and do the data support the conclusions?

Reviewer #1: Yes

Reviewer #2: Yes

3. Has the statistical analysis been performed appropriately and rigorously? 

Reviewer #1: Yes

Reviewer #2: Yes

4. Have the authors made all data underlying the findings in their manuscript fully available?

Reviewer #1: Yes

Reviewer #2: Yes

5. Is the manuscript presented in an intelligible fashion and written in standard English?

Reviewer #1: Yes

Reviewer #2: Yes

6. Review Comments to the Author

Reviewer #1: Thank You for improving the manuscript. I can see the changes made. The sustainability issues are very significant.

Reviewer #2: Dear Author,

All comments have been addressed. Citation and method description arrangements have been made.

Thanks and good luck.

7. PLOS authors have the option to publish the peer review history of their article (what does this mean?). If published, this will include your full peer review and any attached files.

Reviewer #1: **Yes: **Sergey Barykin

Reviewer #2: **Yes: **M. Esra Atukalp

---

## [Author Response · Author response to Decision Letter 1]

16 Nov 2023

Dear Editors and Reviewers，

Thank you for your letter and the reviewer's comments concerning our manuscript entitled " A Study on the Measurement of Inter-provincial Trade Costs in Yangtze River Delta from the Perspective of Value Added Trade and its Promotion Effect" (ID: PONE-D-23-25349). Those comments are valuable and helpful for revising and improving our paper and the critical guiding significance of our research. We have studied the words carefully and have made a correction, which we hope meets with approval. The significant modifications in the paper and the response to the reviewer's comments are as follows:

Responds to the reviewer's comments:

1. Response to comment: The authors have already added the limitations of the research in the conclusion section, but it needs to be brief and specific as some explanations can be part of the discussion section. I would suggest that the authors provide specific key points and recommendations briefly in the conclusion section.

Response: We are very sorry for our negligence and have rewritten this part according to the reviewer's suggestion. 

2. Response to comment: Although the authors have addressed grammatical errors and enhanced the language of the article, it still needs proofreading and editing.

Response: We are very sorry for our negligence and have recheck all the paper according to the reviewer's suggestion. 

Special thanks to you for your good comments. We appreciate the Editors/Reviewers' warm work earnestly and hope the correction will be approved. And also, we want to know the decision earlier if possible because we need this paper for our institutional promotion.

Once again, thank you very much for your comments and suggestions.

Best wishes!

Dr. Yang Wang

Shanghai Dianji University, China

---

## [Editor Report · Decision Letter 2]

23 Nov 2023

A Study on the Measurement of Inter-provincial Trade Costs in Yangtze River Delta from the Perspective of Value-Added Trade and its Promotion Effect

PONE-D-23-25349R2

Dear Dr. WANG,

We’re pleased to inform you that your manuscript has been judged scientifically suitable for publication and will be formally accepted for publication once it meets all outstanding technical requirements.

Kind regards,

Imran Ur Rahman, Ph.D

Academic Editor

PLOS ONE

Additional Editor Comments (optional):

All comments have been addressed.
---

## [Editor Report · Acceptance letter]

30 Nov 2023

PONE-D-23-25349R2 

A Study on the Measurement of Inter-provincial Trade Costs in Yangtze River Delta from the Perspective of Value-Added Trade and its Promotion Effect 

Dear Dr. Wang:

I'm pleased to inform you that your manuscript has been deemed suitable for publication in PLOS ONE. Congratulations! Your manuscript is now with our production department. 

Kind regards, 

on behalf of

Dr. Imran Ur Rahman 

Academic Editor

PLOS ONE